# Brine residues and organics in the Urvara basin on Ceres

A. Nathues [1✉], M. Hoffmann[1], N. Schmedemann[2], R. Sarkar[1], G. Thangjam [3], K. Mengel[1], J. Hernandez [1], H. Hiesinger [2] & J. H. Pasckert [2]

Ceres is a partially differentiated dwarf planet, as confirmed by NASA's Dawn mission. The Urvara basin (diameter ~170 km) is its third-largest impact feature, enabling insights into the cerean crust. Urvara's geology and mineralogy suggest a potential brine layer at the crust-mantle transition. Here we report new findings that help in understanding the structure and composition of the cerean crust. These results were derived by using the highest-resolution Framing Camera images acquired by Dawn at Ceres. Unexpectedly, we found meter-scale concentrated exposures of bright material (salts) along the crater's upper central ridge, which originate from an enormous depth, possibly from a deep-seated brine or salt reservoir. An extended resurfacing modified the southern floor ~100 Myr after crater formation (~250 Myr), long after the dissipation of the impact-generated heat. In this resurfaced area, one floor scarp shows a granular flow pattern of bright material, showing spectra consistent with the presence of organic material, the first such finding on Ceres beyond the vast Ernutet area. Our results strengthen the hypothesis that Ceres is and has been a geologically active world even in recent epochs, with salts and organic-rich material playing a major role in its evolution.

[1] Max Planck Institute for Solar System Research, Justus-von-Liebig-Weg 3, 37077 Goettingen, Germany. [2] Institut für Planetologie, WWU Münster, Münster, Germany. [3] School of Earth and Planetary Sciences, National Institute of Science Education and Research, NISER, HBNI, Bhubaneswar, India. ✉email: nathues@mps.mpg.de

The dwarf planet Ceres is the largest (~940 km diameter) object in the main asteroid belt, orbiting the Sun at a mean distance of ~2.8 astronomical units. Ceres is a survivor of the earliest period of Solar System formation and thus detailed knowledge about its interior provides fundamental insights into the formation and evolution of volatile-rich planetary embryos that originated within the protoplanetary disc[1]. Findings of the Dawn mission[2] suggest that Ceres' interior consists of a thin lag deposit (regolith) layer, an icy crust (density[3] ~1.3 g/cm$^3$) containing the bulk of an ancient ocean, a relict brine layer at the crust-mantle transition, and a lower mantle (<100 km depth, density ~2.4 g/cm$^3$) that is presumably dry and coherent[4]. A denser core, if present, could not be sensed by the gravity data[5].

The Framing Camera[6] (FC) and the Visible and Infrared Spectrometer[7] (VIR) of the Dawn mission revealed local evolutionary processes in surprisingly recent epochs, for example at the Occator crater, where cryovolcanism shaped the cerean surface[8,9]. Less apparent are similar indicators at older sites, which are more affected by erosion, surface relaxation and ejecta blanketing. However, it is suggested that among larger impacts, Occator-like late activity was prevalent and is possibly still traceable through the analysis of the highest spatial resolution FC imagery. The FC imaged the cerean surface in seven colour filters and one clear filter (0.4–1.0 μm), leading to a global colour filter mapping at a pixel scale of ~140 m during the High Altitude Mapping Orbit (HAMO) and at a ~35 m pixel scale for the clear filter during the Low Altitude Mapping Orbit (LAMO)[10,11]. In addition, during its extended mission phase 2 (XM2), clear filter images of Urvara were obtained with a pixel scale down to ~3 m, and these are analysed here for the first time. From the VIR, we added infrared (IR) spectral information obtained in the wavelength range 1.0–4.1 μm at a spatial scale of ~95 m from the LAMO. Details about the data acquisition and calibration can be found under Methods ('Data processing').

In this work, we present a detailed analysis of high-resolution imaging data of the Urvara basin. We find metre-scale exposures of bright material (salts) along its upper central ridge, which originate possibly from a deep-seated brine or salt reservoir. The southern floor shows an extended resurfacing of about 100 Myr after the impact (~250 Myr before today), which is long after the dissipation of the impact-generated heat. In this younger area, one-floor scarp exhibits a granular bright material flow that shows spectra consistent with the presence of organic-rich material. This is the first finding of organic-rich material outside the Ernutet area and its surroundings. Our results strengthen that Ceres is a geologically active world where salts and organic-rich material play a major evolutional role.

## Results

**Geology of Urvara basin.** One of the largest impact structures on Ceres is the Urvara basin at 45.66°S/249.24°E (Fig. 1a, b), located at an average altitude of about −6100 m below the reference ellipsoid[12], and thus it is one of the lowest-lying surface areas. It may have formed within a larger, ancient eroded basin that had a north-south extent of ~240 km. This ancient basin is indicated by a discontinuous circle of elevations, which resemble a second, heavily eroded wall (Supplementary Fig. 1). The older Yalode basin (Fig. 1a) to Urvara's east has likely also influenced its pre-impact geology.

In general, Urvara's morphology is consistent with a complex, medium-aged impact crater, exhibiting a preserved ejecta blanket to its west. The basin exhibits a sharply defined continuous crater wall with vast terraces, caused by post-impact inward slumping, in its south, while a steep single scarp defines the northern and north-eastern rim (Fig. 1b). The wall shows a significant height variation (Supplementary Fig. 1). The sharpness of the northern wall gradually declines towards the east, allowing it to be overtopped by 'smooth material' (SM). Numerous small craters pepper the SM, but larger relief is missing. The origin of the SM at Urvara is important, since it is one of the major geologic units on Ceres, and is younger than the crater[13]. Urvara's location at the transition between the smoother equatorial zone and rougher higher latitudes likely influenced the different morphology of the northern and southern crater walls. The differences in topographic roughness on a global scale indicate the diversity of conditions during the evolution of Ceres' surface. Detailed descriptions of the geology and mineralogy of Urvara using lower-resolution images already exist[13–16], hence the present paper focuses on observations of brine residues using high-resolution images.

A central ridge is located off-centre (Fig. 1b) and rises ~3 km above the southern floor (Supplementary Fig. 2). The ridge marks the highest part of an extended topographic step, separating the elevated northern from the southern floor. The northern flank of the central ridge declines gradually, whereas its southern flank is steep (average slopes up to ~75°), exposing fragmented bedrock (cliffs) followed downwards by a smooth sloped talus that is peppered by numerous boulders (Fig. 2). The cliffs consist mainly of indurated blocky material, extending 300 m below the crest, and exhibit inclusions of bright material (BM) indicative of salts that originate from great depth. In addition, very coarse meso-boulder[17] production proximal to the cliff faces occurs. Many of these boulders tumbled down the talus slope (leaving trails) and accumulated at the base of the central ridge. The presence of meso-boulders and talus suggests that the cliff-forming material is compositionally heterogeneous. The meso-boulders represent a mechanically stronger component that can survive attrition as they tumble down the slope, while the talus represents a mechanically weaker component that erodes away with ease into finer-grained particles and may serve as a matrix material. The mechanical strength of the boulders points to either a certain degree of cementation or their inherent strength. The nature of these cliffs is of importance since they exhibit inclusions of BM possibly originating from a deep source.

Urvara crater contains a central ridge. Central ridges are common features of larger cerean craters, with prominent examples being found in Haulani and Achita. Central ridges are expected to form due to oblique impacts[18,19] or, alternatively, they could be remnants of collapsed central peaks. The orientation of Urvara's central ridge may indicate an oblique impact from the south-west[19]. However, the different morphology between the north-east and south-west crater wall is due to differences in the crustal properties between the southern and equatorial latitudes rather than being caused by an oblique impact, since Yalode shows a similar difference. Moreover, the ejecta field appears symmetrical[13], not showing 'forbidden zones' as would be expected in the case of an oblique impact[18]. Furthermore, the circular outline of the crater wall is improbable for very oblique impacts[20]. Therefore, we conclude that Urvara's central ridge is likely the remnant of a collapsed central peak and is hence expected to uplift and expose materials from the deep interior.

Eastward of the central ridge, we identified a central pit (~20 km diameter, 0.56 km deep) in the basin's average topographic profile (Fig. 3). Central pit craters on Ceres are rare[9] and demand a high volatile content of the crust. The $D_p/D_c$ ratio, in which $D_p$ is the diameter of the central pit and $D_c$ is the (IAU) diameter of the crater, is 0.12. This value fits well within the range of ratios (0.06–0.25) found for cerean pit craters[21]. The model age of the pit (Area 13, see Supplementary Tables 1, 2) is consistent with Urvara's formation age, suggesting a nearly concurrent formation of the crater and the central pit as expected

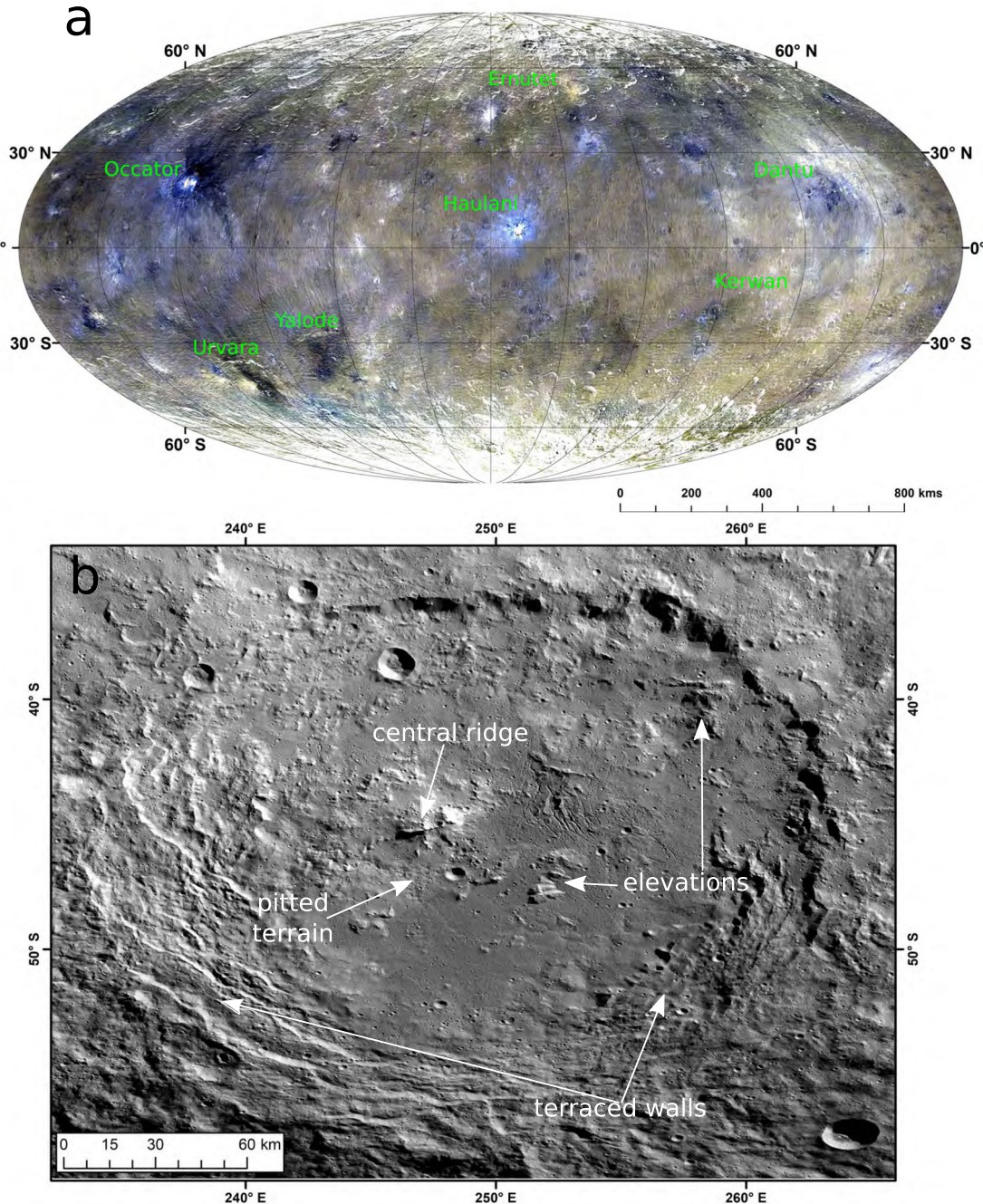

**Fig. 1 Urvara basin. a** Global colour mosaic of Ceres. Locations of craters mentioned in the text are given. Data were obtained from ~1400 km distance during the high altitude mapping orbit and extended mission orbit 3. RGB colours are R = 0.96 μm, G = 0.75 μm and B = 0.44 μm. Mollweide projection centred at the prime meridian. **b** Clear filter mosaic of Urvara seen from low altitude mapping orbit (Framing Camera pixel scale ~35 m). The basin shows all characteristics of a complex crater: central peak/ridge, broad flat shallow crater floor and terraced walls. Important surface features are labelled.

from crater mechanics. Central pits form in complex craters on various planetary bodies but are more frequent on volatile-rich objects, such as the icy moons of the outer Solar System[22]. Central pits are an indication of high volatile content in the subsurface at the time of impact[21,23] and thus are consistent with water ice and the potential presence of brines in the subsurface. Barlow[21] favours a combination of the 'central peak collapse' model[24,25] and the 'layered target' model[26] for explaining the development of central floor pits. Pit formation occurs, according to the first hypothesis, via the collapse of a central peak into a void created by the gradual release of impact-induced volatiles or the drainage of impact melt. The second hypothesis adds the importance of a layered target surface, a stronger surface over a weaker substrate. The global stratigraphy of Ceres matches a stronger surface (crust) that overlays a weaker mantle[27] and thus would be consistent with the second hypothesis. A combination of both these hypotheses could also explain the formation of Urvara's central pit. Thus, the presence of a central pit supports our hypothesis of a partially collapsed central peak capable of revealing subsurface materials.

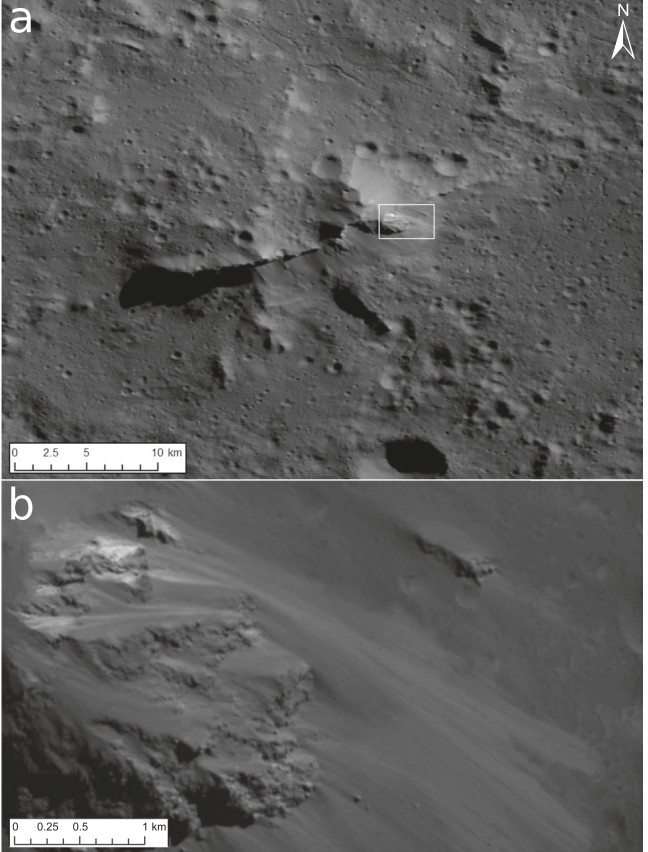

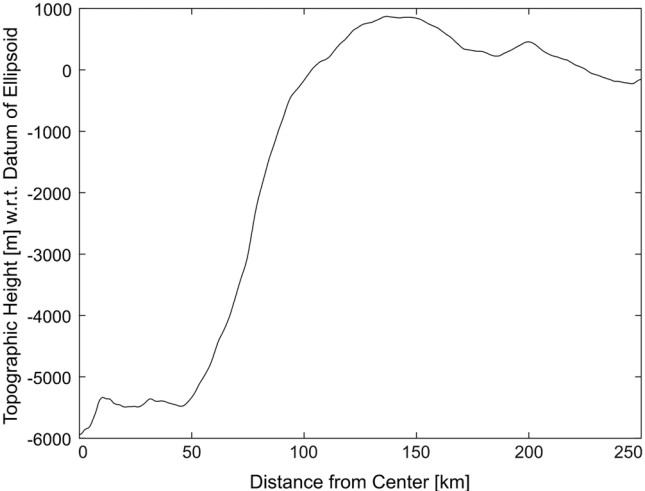

**Fig. 3 Average topographic profile of Urvara.** Profile computed from 360 profiles between the centre of the crater and a distance of 248 km, each offset by 1°. Measurements have been performed using the global high altitude mapping orbit shape model[57]. The central pit extends out to ~10 km distance, the crater wall starts at ~50 km, and a potential wall of an ancient basin is located at ~200 km distance. The average profile leads to a slightly lower basin diameter of 154 km compared to the official International Astronomical Union diameter of 170 km.

**Fig. 2 Central ridge in a clear filter. a** Mosaic at a pixel scale of ~3–5 m from the extended mission phase 2, showing the entire central ridge. It extends ~26 km from south-east to north-west, while the lateral axis measures only ~13 km. The ridge exhibits multiple bright granular flows, which originate from small bright exposures (Central Ridge Exposures) exposed by mass wasting. The box in (**a**) marks the location of the subpanel (b). **b** Extraction of Framing Camera image #0097474 obtained on July 24, 2018 with a pixel scale of ~5 m. The bright material (apparent in the upper left) is concentrated in small nearly point sources, which are in the order of the image resolution. These are the brightest sites in Urvara.

Except for some portions of the northern floor, which display a rugged topography (see geologic map[13]), the floor is dominated by units of SM, whereas in the north and north-east SM even extends across and beyond the crater rim. The SM also appears to blanket older floor units and terraces. Isolated elevations protrude through the SM, especially in the eastern half of the floor. The elevations south-east of the ridge could also be remnants of a former central peak or rough original floor material, which were embayed by impact slurries. Knobby elevations, bay-like features at the outermost contacts to the crater wall and even inside the floor resemble buried craters, which were formed by flow fronts of solidifying impact slurries.

Several linear grooves traverse the eastern and western floor (see Supplement), which could be indicative of post-impact horizontal and vertical movements of the SM and thus could suggest late floor activity, possibly caused by cryovolcanism, late outgassing or/and severe mass wasting.

One of the deepest areas in Urvara is an undulated, irregular unit south of the central ridge (Supplementary Fig. 2), characterised by numerous shallow depressions described as a 'pitted texture'[14]. Some of these depressions (≤2 km) are only partially developed and show no ejecta or elevated rims akin to impact craters. This terrain indicates the past outgassing of a volatile-rich

subsurface, which is in line with the appearance of a central pit. However, flow features[28] that could be indicative of volatiles have not been found.

**Distinctive colour and spectral features**. The colour variation of Ceres' surface proved to be significantly stronger than expected, in terms of both reflectance and spectral shape. For example, young craters often show deposits of BM but also display significantly darker exposures than average (e.g. ejecta)[11]. This spectral diversity demonstrates that Ceres is not a simple agglomeration of materials of uniform composition, but has undergone complex evolutionary processing. Local occurrences of BM are globally present on Ceres. All of these BMs consist of salts as hyperspectral investigations show[29]. Due to their characteristic spectral shape in the FC wavelengths, salty BM can also be identified by using colour images. The bright material on Ceres is often associated with impact craters[30], but some instances exist where it is associated with mons[31] or volcanic features[9]. Understanding the nature, occurrence and distribution of BM on Ceres enables us to draw conclusions on a deep global brine layer, which bears key implications for the cerean evolution models.

Urvara shows a distinctive spectral diversity, much greater than the neighbouring Yalode basin. Figure 4 displays a newly calibrated and processed false-colour mosaic (RGB), with a considerably improved photometric correction of topography. Several colour units can be distinguished. Increased content of BM, whose reflectance is defined as being larger than 0.035 at 0.55 μm[32], raises the overall reflectance of the central ridge area, the rugged terrain and extends beyond the western crater rim. Material darker than Ceres' average is mainly restricted to the eastern part of the floor.

The dark bluish material has a relative reflectance higher than the cerean average at 0.44 μm (Fig. 4, units (4) and (5)). This material often shows a reflectance peak at 0.55 μm, is associated with a number of small impact craters, and is found on the SM as well as Urvara ejecta.

The comparatively high reflectance at 0.44 μm seems linked with freshly exposed material since all these craters appear

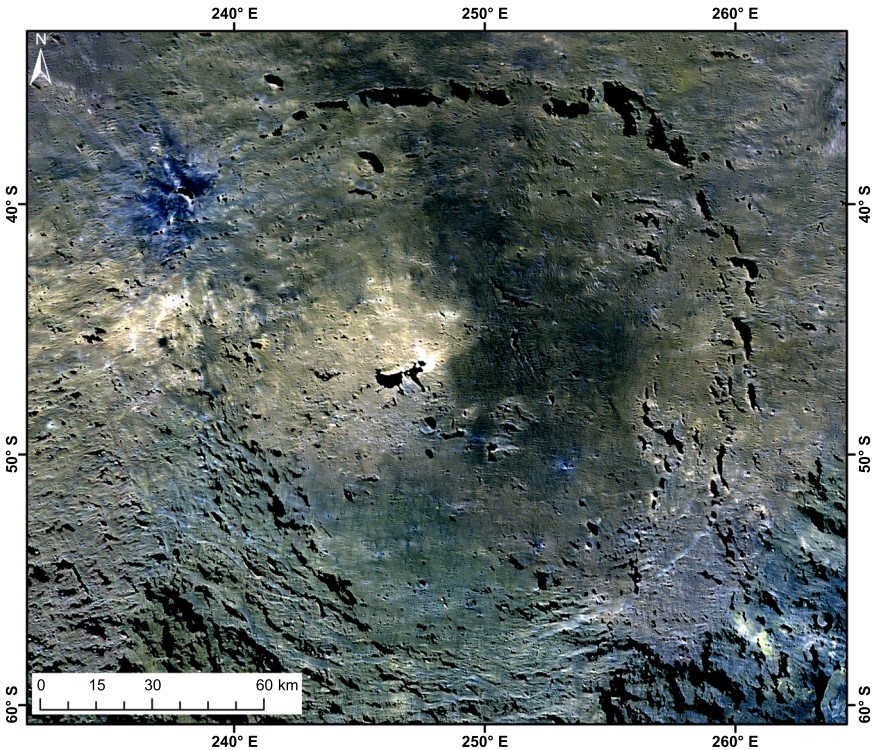

**Fig. 4 Photometrically corrected colour mosaic of Urvara (R = 0.96 μm, G = 0.75 μm and B = 0.44 μm).** This RGB mosaic is in stereographic projection and computed from Framing Camera high altitude mapping orbit data (pixel scale ~140 m) by using the low altitude mapping orbit-based shape model[59]. The following colour units can be identified: (1) 'Central ridge area': showing the brightest ochre-coloured tones and the highest reflectances; (2) 'Rugged terrain': on the western floor in ochre-coloured tones, which expand to the western crater rim, showing slightly higher reflectances than the remaining floor; (3) 'Smooth material': mainly on the eastern floor, which is in Framing Camera spectral data similar to the average dark material of Ceres; (4) A dark bluish crater of 8 km diameter on the north-western rim; (5) Several small bluish-coloured craters showing low reflectances, found on the smooth material and outside the rim and (6) Several small bright material units on the northern crater wall, the floor, the upper central ridge and one terrace.

morphologically uneroded and unit (4) is young (see the section 'Crater-based model ages'). However, dark bluish craters also exist on old SM. Thus, their bluish appearance may relate to ejecta freshness and physical regolith parameters rather than the composition and absolute age of the geologic unit in which they formed.

Spectrally, the younger and older SM (see the section 'Crater-based model ages') appear similar at IR wavelengths. The smooth material is found at many other localities on Ceres, is associated with larger impact craters and is the equivalent to solidified impact melt[33,34], although its granularity and depth are unknown.

Exposures of BM, characterised by their isolated occurrences and sharp contrast with their surroundings (Fig. 2 and Supplementary Fig. 3), display reflectances up to ~0.08 at 0.55 μm and at a ~140 m pixel scale. These exposures have been detected on the upper central ridge of Urvara. We refer to these exposures as 'CREs' (Central Ridge Exposures). The brightest CREs are associated with a cliff at the north-east central ridge (Fig. 2b). They exhibit colour spectra similar to the average cerean BM (see 'Urvara CRE (#1)' spectrum in Fig. 5). The CREs' colour spectra (Fig. 6) differ from those of Occator's faculae, suggesting different salt compositions. Significantly higher reflectances have been measured among XM2 data at pixel scales <5 m. Metre-scale concentrations of BM, associated with a cliff on the north-east central ridge (Fig. 2b and Supplementary Fig. 3), exhibit reflectances up to ~0.16 in the clear filter; these are comparable to those measured for Occator's Vinalia Faculae, indicating high concentrations of salts. Further CREs are found along the ridge, and all are exposed by slumping, mainly on the southern ridge scarp, excavating lenses of BM, which mixes with darker material while moving downslope.

In order to test either common or unique nature of the CREs, we performed a survey of all central peak/ridge craters on Ceres. In total, 239 potential or secure central peak/ridge craters have been identified, but only about 16 of these show BM on their central peaks/ridges at a ~35 m pixel scale. Exposures of BM on central peaks/ridges were found at craters larger than ~25 km in diameter, with Urvara being the largest. Thus, the occurrence of BM on central peaks/ridges for craters >25 km seems to be rather independent of crater size and therefore of the local depth of excavation. Bright exposures on central peaks/ridges are found over a wide range of ages, from young craters like Haulani (~1.96 Myr, LDM)[35] to Urvara, which is more than 100 times older. However, their detectability tends to decay with age, possibly due to an alteration of brine residues as they are exposed to the space environment[30].

Setting a minimum reflectance threshold of 0.05 (at 0.55 μm) we find additional outcrops of BM on the northern and north-western upper crater wall, also exposed by slumping. However, the extent of their total area is relatively small compared to other cerean craters, like, for example, Haulani[36]. The colour spectra of Urvara's bright wall exposures (not shown) reveal that most of these comprise only a minor content of BM, since their spectral shapes are similar to those of the dark background material, but with a slightly higher overall reflectance. Only a few BM wall exposures exhibit the typical reflectance peak at 0.55 μm that is characteristic of cerean BM. A prominent example is found on the eastern crater rim at 40.05°S/260.41°E ('BM terrace (#4)', Fig. 5a). Here, a number of bright dots pepper the upper scarp of a terrace, possibly exposed or deposited by a small impact. A further site is found on the western rim at 42.19°S/237.08°E, where a crater of 1.1 km diameter exposes BM with a reflectance

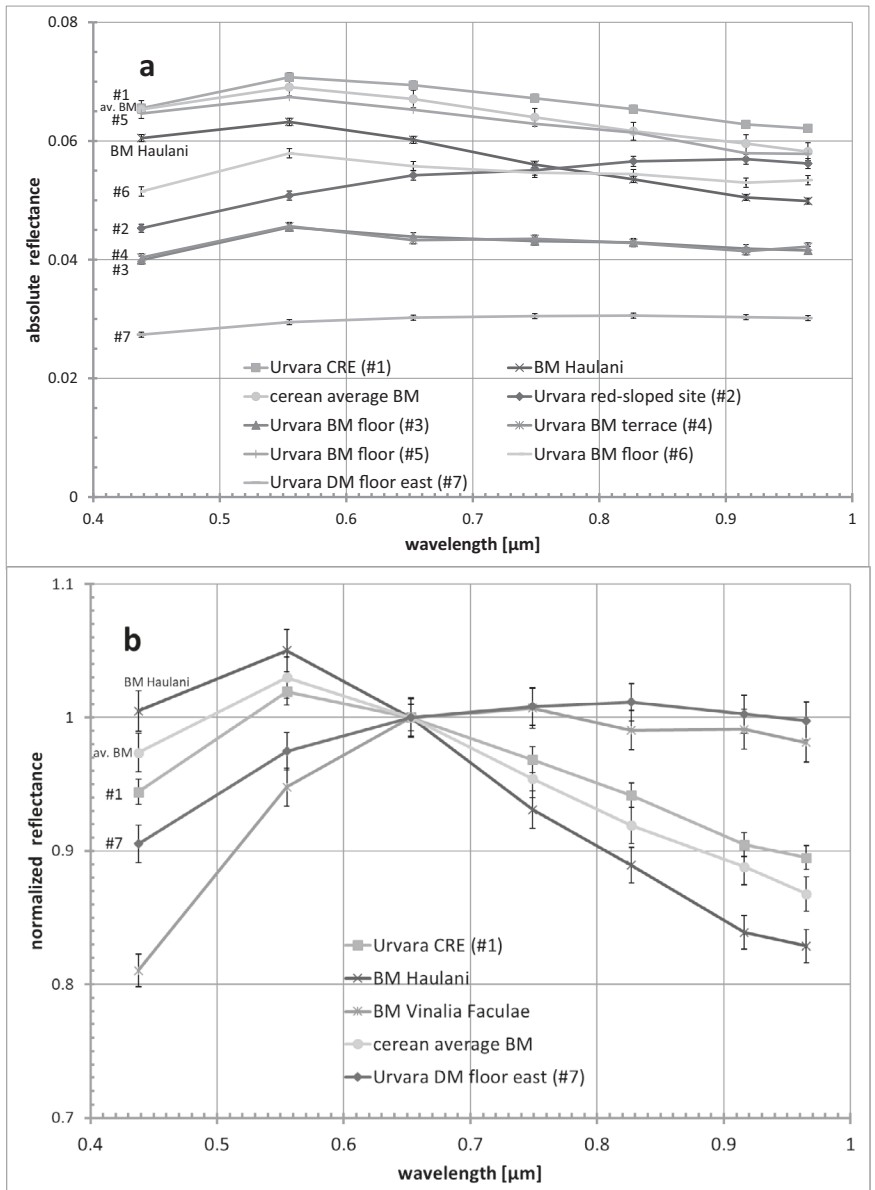

**Fig. 5 Framing camera colour spectra of selected bright material (BM) sites. a** Absolute colour spectra of several BM sites and a dark material floor site. See the text for discussion (for the locations of the sites in Urvara see Fig. 10). All BM sites, except the reddish site, show similar colour spectra at different reflectance levels. **b** Relative spectra, normalised at 0.65 μm, of selected BM sites and a dark floor site. The colour spectrum of the central ridge exposure (CRE) is similar to the cerean average BM spectrum[32], but shows lower (relative) reflectance variations than the BM of Haulani. The CRE spectrum deviates significantly from colour spectra of the Occator faculae (cp. Vinalia Faculae spectrum). Error bars, ±1.5% (typical uncertainty, for explanation, see 'Methods').

of up to 0.07 (at 0.55 μm and at a ~140 m pixel scale). A few small BM sites are also found on Urvara's floor. Most of them exhibit colour spectra, similar to the dark floor but with higher reflectances. However, some of these units show colour spectra similar to the average BM spectrum of Ceres (Fig. 5). Examples are located at 49.18°S/244.36°E ('BM floor (#6)', Fig. 5a) and at 42.19°S/237.06°E ('BM floor (#5)', Fig. 5a). In both cases, craters ≤ ~0.75 km diameter are associated with these deposits. Another site of ~70 m in size is located on the north-eastern floor (38.41°S/255.42°E, 'BM floor (#3)', Fig. 5a), showing a reflectance of 0.044 (at 0.55 μm and at a ~140 m pixel scale). The highest spatially resolved image shows a tiny bright patch, concentrated in a single FC pixel (pixel scale ~33 m) with some brightness-enhanced radially deposited material in its vicinity, resembling a rayed crater.

Surprisingly, a site of ~0.1 by 0.5 km in size on the western floor at 46.53°S/243.26°E (Fig. 7) has different spectral characteristics. This site exhibits an overall red-sloped spectrum in the FC wavelength range without the typical reflectance peak at 0.55 μm of cerean BM outside Occator (see 'Urvara red-sloped site (#2)', Figs. 5a, 6) and thus deviates significantly from all other BM in Urvara. As Fig. 6 demonstrates, the site shows a colour spectrum that is similar to those of organic-rich material found in and around the Ernutet crater[37], but differs from the colour spectra of Cerealia Tholus[9,38]. High-spatial-resolution XM2 imagery reveals that this spectrally red-sloped (reddish) site is located on a scarp on which the BM moves downslope (Fig. 7). This scarp is part of the rugged terrain bordering the young smooth floor material. The BM seems to discharge from the outer (slumping) wall of a small impact crater of ~100 m diameter. The reflectance of the

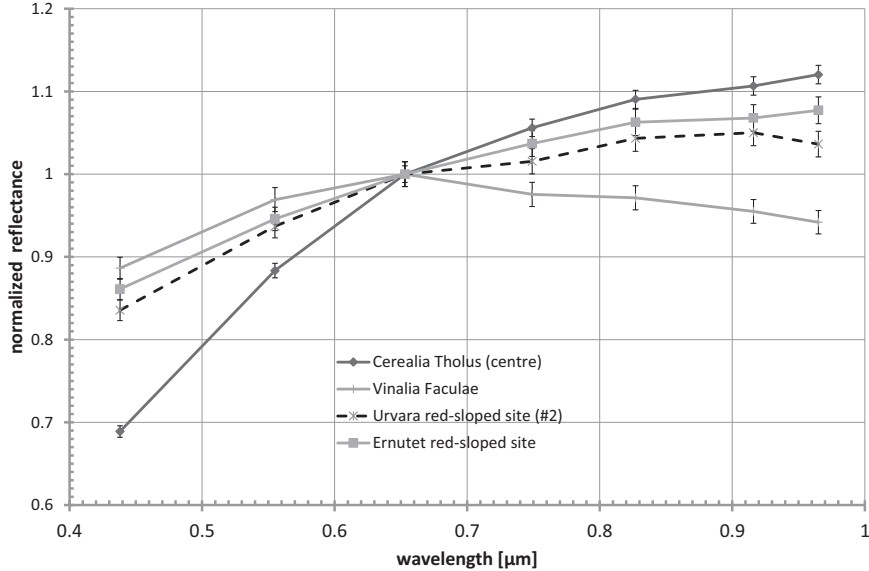

**Fig. 6 Normalised colour spectrum of the bright material at 46.53°S/243.26°E (dashed line).** This spectrally reddish site is located on a scarp of the rugged terrain (Fig. 7) and exhibits a colour spectrum that is similar to the organic-rich material at the Ernutet crater ('Ernutet red-sloped site'). The reddish site exhibits similar and higher reflectances (up to 0.051 at 0.55 μm and high altitude mapping orbit resolution) than Ernutet (typically 0.035). The reddish Urvara site deviates in colours significantly from the faculae material in Occator (Cerealia Tholus and Vinalia Faculae). Error bars, ±1.5% (typical uncertainty).

reddish material reaches 0.13 in the clear filter at a pixel scale of ~3–5 m, i.e. its reflectance is similar to the lower reflectances measured for Occator's faculae but higher than the reddish sites associated with Ernutet, which exhibit typical reflectances of ~0.035. Further BM exposures on other nearby scarps exhibit typical average cerean BM spectra, i.e. they show a reflectance peak at 0.55 μm but no spectral reddening along the FC spectrum. The question arose as to whether the reddish BM contains organic material as the color spectra suggest. To address this, we compared the VIR IR spectra of the reddish BM in Urvara with the spectra of reddish material at Ernutet and BM at Cerealia Tholus (Fig. 8). In VIR data at a ~0.4 km pixel scale, the reddish BM site is barely resolved, i.e. only a few pixels cover the bright flow. As Fig. 8 demonstrates, the IR spectrum of the reddish BM site deviates significantly from that of Cerealia Tholus. The latter exhibits two prominent absorption features at ~3.5 and ~4.0 μm related to sodium carbonate, which are absent in all the other spectra. Thus, the reddish BM in Urvara is of a different composition to the BM of Cerealia Tholus. Next, we verified whether the compositions of the reddish material at Ernutet and the reddish BM in Urvara could be similar. As Fig. 8 shows, both these spectra exhibit similarities. Both display an absorption feature at ~3.4 μm, although the feature is weaker for the reddish BM in Urvara. A ~3.4 μm feature can be diagnostic for organics if it is stronger than a potential ~3.9 μm absorption, while a stronger ~3.9 μm and a weaker ~3.4 μm are diagnostic for carbonates[39]. Thus, we conclude that Urvara's reddish BM likely contains organics. In addition, we found a distinct but weaker ~3.4 μm feature combined with slightly red-sloped FC spectra in the immediate vicinity of the reddish BM. This could indicate a wider distribution of organics on this scarp.

**Crater-based model ages**. The probability distribution of our measured ages (see also 'CSFD model ages' and 'CSFD areas' under Methods), showing four peaks, is presented in Fig. 9. The thick curve is the sum of all thin curves, which represent the individual ages (Supplementary Tables 1, 2). The broad peak at ~250 Myr represents the formation age of Urvara, in line with

previous results[40,41]. This peak comprises several ages (areas 2 A/B, 3, 4 A/B/C, 7, 8, 9, 16 and 17; Fig. 10 and Supplementary Fig. 4) belonging to different geologic units, including the ejecta. The probability peak exhibits a shoulder towards higher ages that is likely caused by secondary impacts, which pepper in particular the north-eastern part of Urvara. A further well-defined peak at ~160 Myr (formed by areas 1 A/B, 5, 6, 10, 11 background and 15) indicates a prominent resurfacing, affecting a continuous area of smooth material on the southern floor and adjacent areas. This finding is remarkable because one would expect the entire floor to show the same age, which is close or equal to the crater formation age. However, we find a difference of ~100 Myr between the crater formation age and the southern SM. The reliability of the age difference (see also 'Robustness of model ages' under Methods) is supported by flow fronts at the border between the young and older smooth material (Supplementary Fig. 5). Counting Area 10 matches the 'pitted terrain' described in the section 'Geology of Urvara basin'. A peak at ~20 Myr in Fig. 9 belongs to a local resurfacing event affecting Area 11, which otherwise shows a background age of ~170 Myr, in agreement with the adjacent young smooth material. Area 11 corresponds to the talus of the scarp shown in Fig. 7, associated with the reddish site (Figs. 5a, 6). Here resurfacing is likely caused by mass wasting(s) on a steep slope. The well-defined peak at ~8 Myr represents the formation age of the dark bluish crater on the north-western crater rim (see Fig. 4 and previous section).

**Discussion**
Urvara's pre-impact crustal stratigraphy must have been altered by the putative ancient basin and the Yalode impact, reducing its volatile content and influencing the peculiar distribution of floor colours/materials. While geologic units contemporary with the formation of the crater can be attributed to crater mechanics, other units like the pitted terrain, the rugged terrain that hosts the scarp showing the reddish material, and the younger smooth material are influenced by post-impact modifications. The measured time difference of ~100 Myr is much longer than the period required for impact slurries to solidify and impact melt chambers

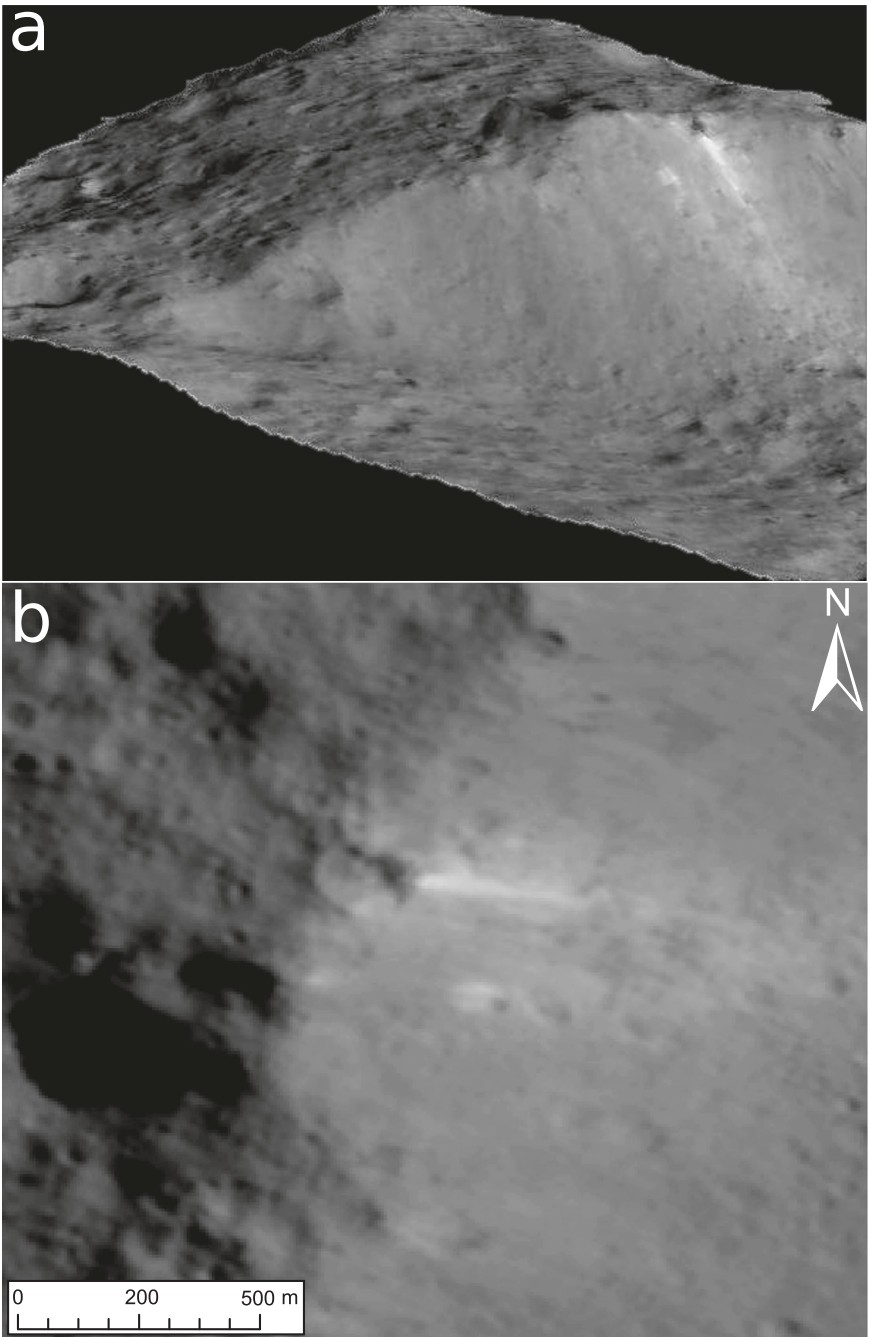

**Fig. 7 Spectrally reddish bright material (BM).** The BM moves downslope on a scarp at 46.53°S/243.26°E (site #2 in Fig. 10). **a** Perspective view was obtained looking to the north-west. Imagery is draped over topography derived from a stereo pair of images from extended mission phase 2 (XM2) orbit (displayed image size is about 3 by 3 km). **b** Clear filter mosaic from XM2 data. An average colour spectrum of this site is shown in Fig. 5a and Fig. 6; its infrared spectrum is presented in Fig. 8. The main source of the BM flow seems to be located at the outer wall of a deformed crater that is located on the upper scarp. Further BM is admixed in its immediate vicinity.

to refreeze. Thus, the question arises as to which process could be responsible for the major resurfacing affecting the floor.

The central ridge exposes the deepest materials in Urvara, probably from a depth down to ~50 km[42,43] below the pre-impact surface. Evidence for such a deep excavation, apart from the large diameter of Urvara, is the enrichment of ammoniated phyllosilicates in the central area[44]. The age of the northern flank of the central ridge (Area 9, Fig. 10) is, within uncertainties, consistent with the crater formation age and thus congruent with both potential hypotheses of origin (see the section 'Geology of the Urvara basin'). Deposition of the CRE material by secondary projectiles at its

exceptional, isolated location is very unlikely; an explanation by an in situ origin is mandatory. Among those, a near-surface origin of the CREs is not likely, since the areas' specific impact history must have led to a severe mixing of materials and a depletion of local crustal volatiles. These findings narrow down the possibilities of CRE origin. One potential explanation is that the BMs were uplifted from great depths when the central peak formed. According to numerical modelling of impact cratering processes[42,43], material from a depth of about 0.5 to 0.7 crater radii can get exposed in a central peak, i.e. the Urvara impact could have reached the potential global brine layer at a depth of ~40 km[9,45–47]. The question is whether frozen or

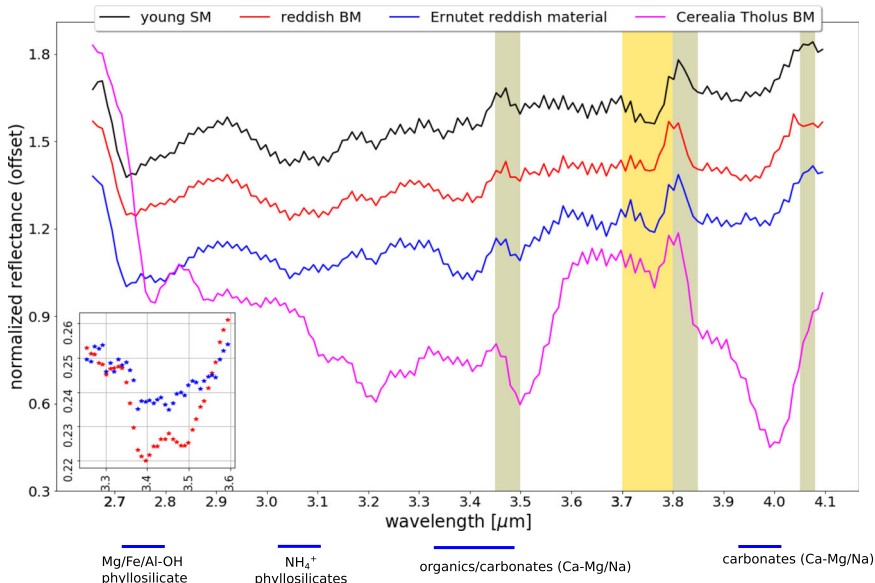

**Fig. 8 Relative infrared (IR) reflectance spectrum of the reddish bright material (BM) in Fig. 7.** The spectrum of this site (VIR cube file m-VIR_IR_1B_1_580121664, XMO6, 2018) is compared with two prominent spectral units on Ceres, which also exhibit red-sloped material in Framing Camera wavelengths. Bars below the x-axis indicate minima locations of typical cerean absorption features and associated materials. While the IR spectrum of the Cerealia Tholus BM (m-VIR_IR_1B_1_521436704, extended low altitude mission orbit, 2016) deviates significantly from Urvara's reddish BM, the spectrum of Ernutet's (VIR_IR_1B_1_498325369, high altitude mapping orbit, 2015) reddish material shows similarities, although the ~3.4 µm absorption feature is weaker for Urvara's reddish BM. The absorption feature at ~3.4 µm is indicative of the presence of organics[39]. This feature is present on the whole scarp (cp. Fig .7) but weakens with distance to the reddish BM and is absent in the nearby young smooth material (SM). The young SM spectrum exhibits typical cerean spectral features: a prominent ~2.7 µm feature due to Mg/Fe-phyllosilicates, a ~3.1 µm feature due to $NH_4^+$ bearing phyllosilicate, and combined ~3.4 and ~3.9 µm features due to Mg/Ca-carbonates[71,72]. The displayed Level 1b VIR spectra have been thermally corrected and smoothened using a Savitsky–Golay filter (3-order), normalised at 3.2 µm, and spectra are offset by 0.15 in reflectance for clarity. The vertical yellow box marks a wavelength range that is heavily influenced by the instrument's order sorting filter. Grey boxes mark wavelength ranges that are considered to be 'artefacts'[71,72]. The inset (lower left) shows the ratio between the young SM and Urvara's reddish BM (blue data values) and the ratio between the young SM and the Ernutet red-sloped site (red data values). The depth of the ~3.4 µm feature varies and is stronger for the reddish BM of the scarp than for the young SM but does not reach the same strength as for areas associated with Ernutet.

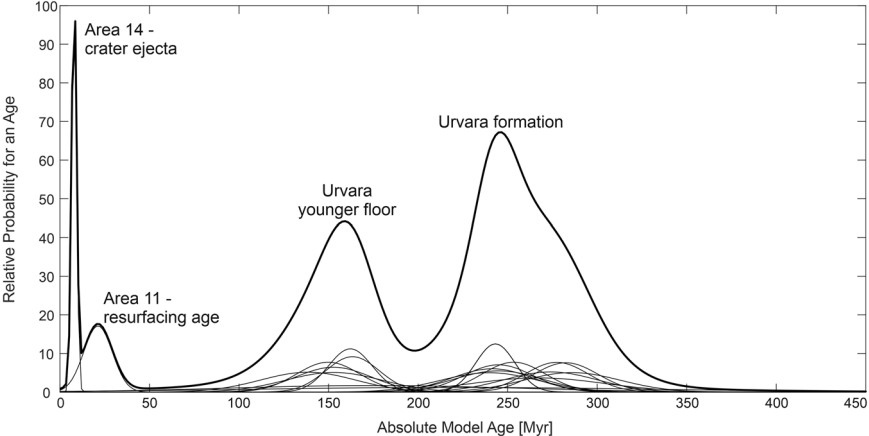

**Fig. 9 Probability plot of crater retention ages.** Displayed results are based on the lunar-derived model (LDM, see 'Methods' and 'Supplement'). The x-axis shows the absolute model age in Myr, while the y-axis displays the relative probability for a given age. The thick curve is the sum of all thin curves, which represent the individual areas measured (for details see text).

liquid brines from the crust-mantle transition would survive an uplift during the central peak formation to the near-surface and would leave behind intact salt lenses within the central peak, since the uplift leads to severe heating, shattering and deformation of the material. An alternative explanation for the origin of CREs is cryovolcanism. The transport of brines from depth could have started when extended crustal cracks formed during the impact or later through crustal shrinking due to the cryomagma chamber

refreezing. These cracks could have connected a deep-seated brine reservoir with the surface. Then the exsolution of gases at depth could have led to the ascent of brines. The presence of gases at depth is inferred from the composition of Occator's faculae, which are rich in sodium carbonate. If brines migrate towards the surface, only two principal possibilities exist for separating the water from the dissolved salts: (1) cryovolcanism: on reaching the surface brines start to boil, water is lost to space immediately, salt crystals form, and are

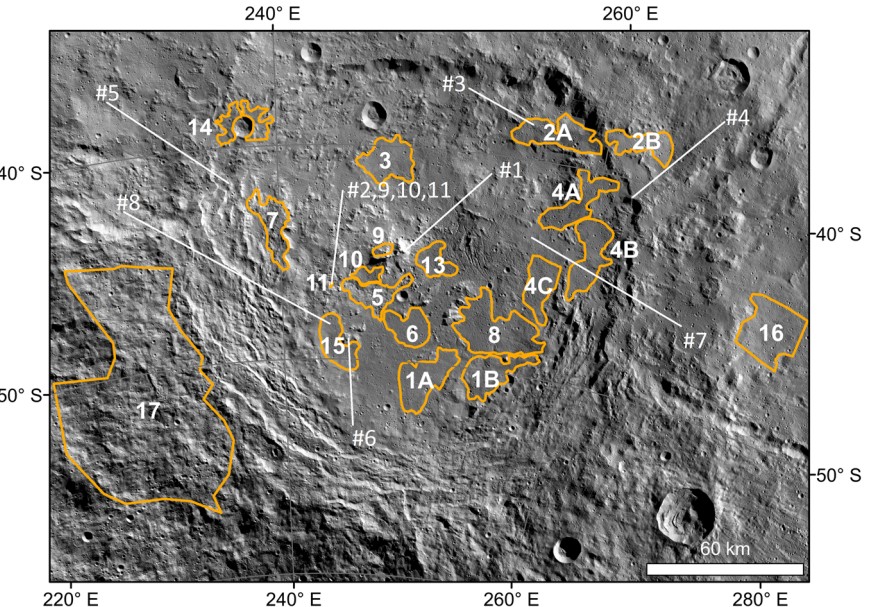

**Fig. 10 Crater size-frequency distribution areas of Urvara.** The location of the crater counting areas (orange outlines), as well as the sites of the used spectra in the manuscript, are shown. The mosaic is in azimuthal equidistant projection centred at Urvara. The exact borders of the counting areas as well as the derived crater counting functions and ages are presented in Tables 1 and 2 of the Supplement.

deposited on the surface; later on they are covered and mixed with darker material; (2) subsurface intrusion: brines do not reach the surface but reside at near-surface depths. Solvent loss is achieved by capillary transport of brines through the porous upper crustal material; brines possibly solidify upon cooling and releasing water into space through sublimation of the icy component leaving behind salts. The contribution of carbon dioxide to such volatile activity cannot be quantified, since there is no corresponding remnant, although the presence of carbonates is a strong relevant indication.

If the proposed mechanisms were truly responsible for the CRE formation, it should also explain the related phenomena in the surrounding floor. Subsurface cavities/porosities filled with brine (and its residues) might have formed the CREs and possibly also subsurface floor BM. Similar processes are described for the Occator crater[9,38,45,47] in which bright extensive salt deposits document cryovolcanic activity, well beyond the epoch of the formation of the crater, and a link between the cerean surface and a deep-seated brine reservoir. However, it remains to be explained why evaporites occur mainly on the central ridge rather than on the floor. The relatively low number of BM exposures on the floor rather tends to negate surface volcanism, but former salt deposits could have been wiped out by severe mixing and irradiation. We did not find spectrally reddish material among the CREs, which could be a consequence of a time-dependent decay of such a spectrally red component (organics) due to the higher age of Urvara. In summary, a cryovolcanic activity that did not reach the surface seems more likely to us for Urvara.

The clearest hint of volatile activity is the presence of sublimation or devolatilization pits on a low-lying floor area (Supplementary Fig. 2). Violent outgassing of the aqueous vapour phase must have occurred ~100 Myr years after crater formation (Area 5, Fig. 10), confirming the former presence of significant volatile content in the subsurface, outlasting the violent impact history. In our interpretation, due to the release of volatiles, the pre-surface of the "pitted terrain" subsided to the current elevation level and caused instabilities and small-scale collapses. Thus, it seems that these volatiles were involved in a resurfacing that completely reset the cratering record in the southern floor. Hence, the observed morphology supports the cryovolcanic interpretation.

Smooth material covers different units (see 'Geology of Urvara basin') and three hypotheses of origin have been suggested[13]: (1) localised extrusions of volatile-rich material; (2) remobilisation of the volatile-rich rim and/or terrace materials; and (3) deposits from Yalode and Urvara that have merged together to form a single unit (fallback material). Hypothesis 1 could explain the formation of the young SM, although no source vents are seen. However, the old SM is spread over different elevations and thus an extrusive origin seems unlikely. Hypothesis 1 also has the difficulty of explaining by which process (energy source) the younger SM could have extruded ~100 Myr after the formation of the crater. Possibly, cryomagma formed and extruded in a similar process to that suggested for Titan[48]. Resurfacing on Titan is explained by occasional cryovolcanism, caused by cracking at the base of the ice shell and formation of liquid pockets in the ice— somehow similar to the predicted situation in the cerean lower crust. Hypothesis 2 would require a late large-scale event, triggering remobilisation of the wall and terrace material. Such an event could be a large near impact, causing instabilities of the crater wall and terraces. However, whether this could lead to extended floor flooding and resurfacing of adjacent elevated areas is highly debatable. No indication of such an outer influence is detectable. Hypothesis 3, in which fallback ejecta form the SM, is inconsistent with ejecta modelling, since only a minor fraction of the ejecta returns to its source crater[49]. Moreover, the younger age of several SM units does exclude the fallback hypothesis. On the other hand, the similar age of the younger SM and the pitted terrain indicate a link and the possibility that both formations could have been caused by one event. However, the case is different for the western floor, where significant amounts of BM are admixed over a wide area. This BM was possibly directly excavated from the lower crust by the impact, whose transient crater must have reached the brine layer at a depth of ~40 km. Again, the multiple instances of BM demonstrate the presence of a brine/salt layer before the impact. Regardless of whether the formation of the CREs occurred due to an uplift process during the central peak formation or through late cryovolcanism, the presence of liquid or frozen brines or their remnants at great depth is required for explanation and supported by evidence. Thus, we conclude that brine/salt layers at different depths exist in the cerean

crust or even form a single layer, which shows a varying depth. However, the intense bombardment of Ceres by large projectiles during its evolution, including the ancient impact mentioned above, could have caused a local thinning, depletion, or even destruction of an ancient deep-seated brine layer, reducing the amount of detectable BM in Urvara.

The presence of organic material in Urvara, the first detection outside the Ernutet area, is enigmatic and significant. Its sensitivity to spectral degradation by space weathering[50] may be responsible for its scarcity. The exposure in Urvara resembles a representation of a similar distribution of the organic material as in the Ernutet region: comparably small in extent and confined to low depth. The organic absorption at 3.4 μm and the red-sloped FC spectra extend to the immediate vicinity of its brightest location. It seems that this brightness represents the purest form of the organics in terms of concentration and its pristine status as a very fine-grained bright deposit. As in Ernutet, there is no obvious sign of a fresh local impact justifying exogenic infall as a source of origin. However, the spectral properties of the reddish material are distinct from those of salty BM. Due to the sensitivity of the organic material to alteration, its presence in the impact-heated and resurfaced floor of Urvara is remarkable. It raises questions on the carbon chemistry of Ceres' crustal or deeper materials and points to a much more widespread distribution during earlier epochs of Ceres' evolution.

Our results on Urvara prove, as for Occator[9], a post-formation activity, causing resurfacing of a significant fraction of Urvara's floor ~100 Myr after crater formation, i.e. long after the dissipation of the impact heat. The organic-rich material on a relatively young floor scarp is independent of an impact origin. In addition, the CREs are hard to explain without liquefaction. However, a deep brine layer may enable resurfacing, fresh salt deposits, and the injection of organic material from below. Our findings strengthen the view of a dwarf planet, which is a recently geologically active world, where salts play a prime role in preserving liquids in a heat-starved body.

## Methods
**Data processing**. Dawn's FC imaged the surface of (1) Ceres from several different orbits between 2015 and 2018. Global mapping in seven bandpass and one clear filter has been achieved from Survey, HAMO and LAMO orbits at pixel scales of ~400, ~140 and ~35 m, respectively, whereby LAMO colour data were restricted to a number of high-priority surface features, excluding Urvara. After completing its primary mission phase at Ceres, Dawn was extended twice (XM1 and XM2 mission phases), leading to further colour and clear filter mapping with often similar spatial resolutions to those during the prime mission. However, in June 2018, the spacecraft entered its final elliptical orbit, enabling spatial resolutions of the Occator and Urvara regions of a magnitude better than before (pixel scale up to ~3 m). This clear filter imagery of Urvara in combination with the colour data of the prime mission (HAMO) of the same region has been analysed in the present work.

The calibration of the FC imagery is detailed in a number of publications and project reports[51,52]. Images exist at different processing levels: 1a (raw), 1b (radiometrically corrected) and 1c (stray light and ghost-image corrected). We used the highest corrected data[51] level 1c, which is converted to reflectance (I/F) by dividing the observed radiance by the solar irradiance from a normally solar-illuminated Lambertian disk. All used mosaics and colour cubes are computed by an extensive USGS ISIS script[45,53], involving a photometric correction to standard viewing geometry (30° incidence, 0° emission angle, and 30° phase angle) using Hapke functions[54]. Hapke light scattering parameters have been iteratively derived from the Approach, Survey, and HAMO mission phases, starting from parameters obtained by ground-based observations[55]. Iteration of these parameters became necessary because increasing image resolution leads to the increasing significance of sub-pixel shadows, which change the weight of contributions from different latitudes to a globally integrated model fit. For example, a transition from a one-parameter Henyey-Greenstein function to a two-parameter description[56] was required for the HAMO data. For the photometric correction and the visualisation of our mosaics, we used the HAMO Ceres shape model deduced from FC clear filter stereo images[57]. Alternatively, especially for the higher-resolution data, we computed stereo digital terrain models (DTMs) from LAMO and XM2 clear filter imagery by using the Ames stereo pipeline[58] or used the LAMO SPC shape model[59]. The resulting reflectance data are map-projected in several steps, and co-registered to align the colour frames, creating mosaics/colour cubes for further analysis.

The uncertainties of the presented reflectances are based on measurements of homogeneous areas in different mission phases and illumination geometries for which various measurement box sizes were analysed[60,61].

**CSFD model ages**. Our approach to estimating absolute surface model ages is based on the analysis of crater size-frequency distribution (CSFD) measurements[62]. The underlying chronology model for our analysis is based on the link between radiometrically dated lunar surface samples brought back by sample return missions (Apollo, Luna) and the measured CSFDs of the related geologic units on the Moon[63–65]. The lunar chronology system has been converted to the impact conditions on other bodies such as Mercury, Mars and main belt asteroids[66]. The conversion from the lunar chronology system to the lunar-derived Vestan chronology system has been tested successfully for the cases of basin formation ages on Vesta and radiometrically dated Vestan meteorites that recorded those energetic events[66]. The cerean chronology system and its relation to the lunar chronology system was detailed previously[67]. In order to derive meaningful ages, it is required to measure crater frequencies on geological units that were formed by a specific process at a specific time[68]. Within a geologic unit, each subarea should show the same result. Uncertainties of model ages are influenced by statistical and parameter uncertainties as well as uncertainties caused by geologic conditions[9].

**Robustness of model ages**. For Occator small differences of ~4 Myr in crater retention ages between its ejecta blanket and its crater floor have been linked to uneven material properties that allowed the same projectiles to create slightly larger craters in the lower-strength ejecta material[9,69]. The same cause has been considered for the measured age difference on the floor of Urvara. In order to explain the observed age difference of about 100 Myr, the craters in the younger unit would need to be about 14% smaller than those craters in the older unit, which is similar to comparable observations on the Moon[70]. However, in contrast to the Moon and Occator crater, in Urvara the age differences occur within the floor of Urvara with lower retention ages constrained to the south-eastern part of the crater floor. It is not immediately clear to us why the material properties should change while the floor morphology remains similar. In fact, the region where the change in crater retention ages occurs shows lobate morphology, implying a quasi-fluid deposition of a layer superposing the existing crater floor. Digital Elevation Models (DEMs) from high-resolution stereo images suggest layer thicknesses of at least a few decametres up to more than 100 m. One example of such a lobate layer is presented in Supplementary Fig. 5. Since none of our CSFD measurements in the younger smooth unit appears to show a resurfacing kink that would support a superposing thin layer erasing only the smaller craters, we conclude that all craters that formed during the ~100 Myr between crater floor formation and the emplacement of the superposing smooth layer were completely erased. From our crater chronology model, we believe that between one and two 1 km craters should have formed in our respective measurement areas during that time, which have been erased. These numbers increase by the power of three, going to smaller crater diameters (N∝D-³).

We mapped craters in Area 10 with increased attention, such that pits are excluded, as they do not bear any age information. Misclassification of pits into craters would lead to an overestimation of the surface age.

**CSFD areas**. Figure 10 presents an overview of all analysed areas, which can roughly be subdivided by their location. Areas 14, 16 and 17 dates the formation of crater ejecta blankets. While Areas 16 and 17 dates the formation of the Urvara ejecta and thus the formation of Urvara itself, Area 14 dates the formation of a dark bluish 8-km-diameter crater on the north-western crater rim of Urvara. Areas 2 A/B, 4 A/B and 7 are located in the terraced area of Urvara. Usually, these terraces form shortly (minutes to days) after the crater formation. Thus, these areas give a minimum age for the formation of Urvara. Areas 1 A/B, 3, 4 C, 5, 6, 8, 9, 10, 11, 13 and 15 represent different sections of the crater floor and can signal post-formation resurfacing events as has been shown in the Occator crater[9]. Area 11 is the smallest area investigated and follows the foot of a scarp in which mass wasting may occur, for instance, due to seismic shaking. Usually, such areas are avoided for surface age determination because of reoccurring mass wasting events, but in this case, our goal was to date an event that may have happened at the scarp in conjunction with the exposure of the spectrally reddish material.

## Data availability
The Framing Camera and VIR data are available through the PDS Small Bodies Node website (https://pds-smallbodies.astro.umd.edu/). Higher data products that support the findings of this study are available from the corresponding author upon reasonable request.

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

## Acknowledgements

We thank the Dawn mission team for the development, cruise, orbital insertion, and operations of the Dawn spacecraft at Ceres. Also, we would like to thank the instrument operations teams at MPS and INAF. The Framing Camera project is financially supported by the Max Planck Society and the German Space Agency, DLR.

## Author contributions

A.N., M.H. and G.T. supported the Dawn operations, leading to the used FC data. A.N., N.S., R.S. and G.T. contributed to the data analysis. The manuscript was written by A.N., M.H., N.S., R.S., K.M. and J.H.P. with detailed reviews and further contributions by H.H. and J.H.

## Competing interests

The authors declare no competing interests.
