## [Peer Review File · Nature Communications]

The Urvara basin on Ceres – brine residues and organicsREVIEWER COMMENTS

Reviewer #1 (Remarks to the Author):

Review of "The Urvara basin on Ceres – brine residues and organics" by Nathues et al.

Summary

The central claim of the paper includes the identification of salts and organic material in the Urvara basin on the dwarf planet Ceres. Assessing the distribution of organic material throughout the solar system is an area of high scientific interest. The finding of organic materials in the Urvara basin is of interest because, to date, the Ernutet area is the only location on Ceres where organic materials have been identified. These results are not entirely novel (organics have been discovered elsewhere on Ceres, for example <https://doi.org/10.1038/nature18290>), but they do offer the possibility that organic material is more widespread on the dwarf planet than previously understood.

Main Comments

My most considerable comment is that, in its current form, the paper does not read as though it is written for the greater planetary science community and the typical audience of a journal like *Nature Communications*. The current presentation may hinder the perceived significance of the authors' findings. Specifically, *Nature Communications* enjoys one of the widest audiences of any Nature-family journal due to its open-access status and science communication focus, and papers submitted here should be widely accessible. I suggest a moderate revision to improve this aspect of the paper.

As a stylistic comment, that paper cannot be fully comprehended without reading it side-by-side with the Methods. Some important details are relegated to the Methods. Elsewhere information and nomenclature are defined within the Methods and given without explanation in the main text. I suggest making the main text more stand-alone so that a reader can understand the paper's central points in a single coherent read, and then can delve deeper into the details via the Methods. Accordingly, more emphasis should be placed on presenting the central results with their greater context. I recommend the authors make the following changes to improve this aspect of the paper:

Please alter Figure 1 to provide more geospatial context and help the reader understand the geologic setting of the findings without constantly referencing the supplement or other manuscripts. My suggestions here are as follows:

- Extended Data Figure 1 shows where on Ceres Urvara basin is located. I would include something similar (but smaller) as an inset in Figure 1, so the reader immediately has geospatial context for the basin. I would also add the Ernutet area to this inset so the reader can see it is in a different hemisphere, thereby strengthening their finding that organics may be widespread. I would suggest adding other craters/basins referred to throughout the paper for context, as well.
- Currently, Figure 1 shows the Urvara basin in both nadir and perspective views; the perspective view does not add much to the figure. I would revise Figure 1 to include a nadir view that is clean (as the current panel a) and one that is annotated (a new panel b).
 - It would be helpful to directly annotate some of the features described in lines 5090 in the suggested new panel b. Otherwise, the reader is left to assume they are correctly identifying the features described in the text on their own.

In lines 50-90, the "Geology of Urvara basin," several physiographic descriptions are provided about Urvara without always explaining the significance of these features. I suggest that for every physiographic description, the authors add a few words to tell the reader *why* this is pertinent to the high-impact results (finding of putative salts and organics). For example, there is a lot of text dedicated to presenting the smooth material in this section—I believe this is relevant because it implies that some resurfacing took place ~100 Myr after the crater's formation, and the area is relatively young. This detail should not be relegated to the Methods. Any initial discussion of the geology should explicitly frame the main results.

In lines 96-110, there is a brief description of the bright material on Ceres. Line 101 states: "These areas show mostly higher reflectances than the smooth material, often in the order of or exceeding the lower limit of BM on Ceres (0.035 at 0.55 μm)¹⁸..."

Please add one sentence to explain the context of other BM on Ceres, where it has been discovered, and what it might be. For example, "BM discovered elsewhere on Ceres, is often associated with impact craters, but some instances exist where it is associated with mons or volcanic dome-like features¹⁸". Good context is provided at the end of the section (line 121) for the very bright material. If the other reflectance units described in this section have analogs elsewhere on Ceres, please provide their context.

In lines 194-201 contain a lot of detail that is not decipherable without referencing the Methods/Supplement. I suggest shortening this section and moving most of the text to the Methods, then distilling the essential results relevant to the paper's main findings.

Aside from the comments on the presentation described above, the work itself is convincing. Several previous color and spectral studies using the Dawn payload have been published and have demonstrated the validity of data returned by those instruments. The statistical analysis on crater ages appears thorough. I suggest where the authors should add additional details in my minor comments below.

Minor Comments

Line 99: Here, the term/unit Bright Material (BM) is mentioned for the first time. Please move up the quantitative definition of BM (currently on line 102) to the sentence where the unit is introduced.

Line 195: Above, I suggest much of this material be moved into the Methods. In the Methods, please explicitly define the criteria used for identifying pits versus craters. On line 986 (Supplement), I see the distinction that pits are "lacking raised rims", but any other criteria used during crater identification/counting should be explicitly stated in Methods.

Line 299: Exposures of (concentrated) BM showing higher reflectivity [insert reflectance threshold here] on floor and ejecta are rare.

Line 312: "The detection of organic material on a relatively young floor scarp let us conclude that organic material could be more widespread on Ceres than thought and in this case would be of endogenic origin and not due to exogenic infall."

I am not confident organic detection is not potentially exogenic in origin, given many recent findings of exogenous material on other asteroids (e.g. Tatsumi et al., 2021, DellaGiustina et al., 2021). Please add another sentence justifying this statement.

Line 430: Please use the author's last name (Reddy) for the citation

Line 586/Figure 3: Please state the pixel scale or ground sample distance in the caption for the mosaic shown.

Line 605/Figure 4: States "Error bars, $\pm 1.5\%$ (typical uncertainty)." Please provide the relevant calibration paper/reference and specify a relative (e.g., filter-to-filter) or absolute uncertainty.

Line 693: "presence of volatiles in the subsurface. Barlow and co-authors"

Line 964: "The grooves are certainly not secondary crater chains, as we do not observe raised rims or ejecta deposits"

Line 967: Supplementary Information Table 2: Crater Plots and Areas of Measurement. Many of the legends in these plots, and some of the plots themselves, are not legible. Please correct this in the next revision.

Reviewer #2 (Remarks to the Author):

General comments:

Although interesting, I find the paper not so easy to read because of the multiple back and forths between main text and the methods sections / extended data figures.

Important information on data analysis is given in the supplementary material, but shouldn't it be in the methods sections?

On the contrary, in the methods there are sections that are not « methodology » but give more description (central ridge cliffs, dark bluish material, nature of central ridge, rising brines): I do not understand the hierarchy between the different sections and kind of text/data/method/figures you present at the several places.

For example, in the section « crater based model ages » you cite ROIs number that are only defined lately in the extended data figures (#8 of 10): it is not easy to follow the main text while looking for important informations farther in the paper. Or, all the sites describes in the « color and spectral features section » has their spectra showed in another extended figure, but isn't it the main purpose of the paper?

There is also some confusion between what is new content in this study and what has been published previously. Especially, line 58-59 you write « Descriptions of the geology and mineralogy of Urvara using lower resolution images can also be found elsewhere ». Thus, is the figure 1 as important to be put in the main text? The same for figure 3 (based on HAMO data), that is in a way presented in Crown et al. 2018 (your ref 12)?

Is the whole description in the section « Geology of Urvara basin » mandatory as it was described previously by other authors?

Also, for the crater-based model ages, some equivalent work has been done by Crown et al.: how do your study and results compare to this publication (your ref 12)?

Another concern is that your paper title refers to mineralogical hints on Urvara, precisely brines and organics, but there is not that much about spectroscopy in here. The description of features is not sufficient to convince the reader: you use a reflectance level to conclude on the presence of salts, and similar colors to conclude on organics.

Most of text and figures is related to geologic context that was already previously discussed.

Specific comments:

- line59: could you remind the reader what was the global mineralogy described by previous study for Urvara?

- I recommend to not use parentheses for words that are incorporated in a sentence. E.g. l.105 « However, the (younger) smooth material » or l. 106 « The (dark) bluish craters ». It is not clear whether these words are hypothesis or important information. There are other occurrences throughout the text, please adopt another convention.

- line215: you use the word « ammoniated » in the discussion/conclusion as a major constituent to explain geologic features, but this is not used before in the text and is included in fig.5 (NH₄) without explanation. Please clarify your spectroscopic detections (link with salts?).

- Fig2: I am not sure I understand what the « cross-axis » is here.

- Fig3: I do not see the difference between the feature related to organics and the one to carbonates in the 3.4 microns region? Are the arrows misleading? Are they highlighting different components (you use « organics », « organic component »)?

Furthermore, what are the spectral signatures around 3. 1 and 3.2 on the bottom spectrum (Cereal Tholus)?

- Fig5: Give in the caption the site number (e.g. red sloped terrain is #2)

Also, you mention « the BM site in fig.7 », but this should be « extended data figure 7 ».

There are no available VIR spectra for the other BM sites you present in ext. data fig. 6 to enrich the discussion?

- line636: should « orbit » be plural?

- line643: should « These » be singular?

- line 648: same for « reflectances »?

- line667: I would define also here in the Methods section what the CREs acronym is (if the reader goes to this section while reading the main text in which the acronym is defined later).

- line739-740: I do not understand the statement « ... to the Moon and Occator crater in Urvara the age differences occur within the floor of Urvara... ». Is there a coma missing after « crater »?

- ext. data fig2: What about including a ROI on this figure instead of the arrow? The arrow is hard to see and the ROI might define better the edge of the following figure 3.

- ext. data fig3: Could you give the same longitude than on the previous figure2 (use E instead of W)? It could be easier for the reader to localize the region.

- ext. data fig4: the x-axis is a bit weird, is there a reason that you give the numbers 99, 149, 198, and 248 instead of 100, 150, ...?

- ext. data fig5: Is it possible to put a scale on both views? And to separate them (a vertical white line or something) to see at first sight these are two distinct images (and not one view with some « blank » spaces on the FOV)?

- ext. data fig6: I find these two plots hard to read (especially panel A because of all the plotted data). Could you add a number/letter/sign in front of each line (around 0.4 microns) to guide the eye?

- ext. data fig7: specify in the caption this is site #2 to remind the reader with previous cited sites.

- ext. data fig9: there is no explanation in the caption, is it intended? What about put this figure at the same scale than the previous one fig8 ? Or put the numbers from fig8 ROIs to correlate both figures? Just a question (might be out of scope, my curiosity), but what could be the age of all the uncolored areas?

- ext. data fig10: It is hard to see the different regions you cite. Could you add some lines/arrow/signs to point towards what you describe in the caption?

Review of “The Urvara basin on Ceres – brine residues and organics” by Nathues et al.

Summary

The central claim of the paper includes the identification of salts and organic material in the Urvara basin on the dwarf planet Ceres. Assessing the distribution of organic material throughout the solar system is an area of high scientific interest. The finding of organic materials in the Urvara basin is of interest because, to date, the Ernutet area is the only location on Ceres where organic materials have been identified. These results are not entirely novel (organics have been discovered elsewhere on Ceres, for example <https://doi.org/10.1038/nature18290>), but they do offer the possibility that organic material is more widespread on the dwarf planet than previously understood.

Main Comments

My most considerable comment is that, in its current form, the paper does not read as though it is written for the greater planetary science community and the typical audience of a journal like *Nature Communications*. The current presentation may hinder the perceived significance of the authors' findings. Specifically, *Nature Communications* enjoys one of the widest audiences of any Nature-family journal due to its open-access status and science communication focus, and papers submitted here should be widely accessible. I suggest a moderate revision to improve this aspect of the paper.

>> You are completely right. The manuscript was originally written for and adapted to the style of *Nature Astronomy* and thus had to comply to a specific (lower) word limit for the main text. However, the updated version (updates in bold font in the manuscript) is now restructured to fit with *Nature Communications*.

As a stylistic comment, that paper cannot be fully comprehended without reading it side-by-side with the Methods. Some important details are relegated to the Methods. Elsewhere information and nomenclature are defined within the Methods and given without explanation in the main text. I suggest making the main text more stand-alone so that a reader can understand the paper's central points in a single coherent read, and then can delve deeper into the details via the Methods. Accordingly, more emphasis should be placed on presenting the central results with their greater context I recommend the authors make the following changes to improve this aspect of the paper:

>> The entire text is restructured. The main text now contains all essential information. We would like to thank you very much the fruitful comments, suggestions, and your time spent on the manuscript!

Please alter Figure 1 to provide more geospatial context and help the reader understand the geologic setting of the findings without constantly referencing the supplement or other manuscripts. My suggestions here are as follows:

- Extended Data Figure 1 shows where on Ceres Urvara basin is located. I would include something similar (but smaller) as an inset in Figure 1, so the reader immediately has geospatial context for the basin. I would also add the Ernutet area to this inset so the

reader can see it is in a different hemisphere, thereby strengthening their finding that organics may be widespread. I would suggest adding other craters/basins referred to throughout the paper for context, as well. >> **Ok, thanks, we have now combined the geospatial context figure with the overview figure of Urvara basin. We tried an inset but its readability was insufficient.**

- Currently, Figure 1 shows the Urvara basin in both nadir and perspective views; the perspective view does not add much to the figure. I would revise Figure 1 to include a nadir view that is clean (as the current panel a) and one that is annotated (a new panel b). >> **We deleted the perspective view as you suggested and added the geospatial context instead. These new subfigures (a) and (b) show the major features of the cerean surface and of Urvara.**
 - It would be helpful to directly annotate some of the features described in lines 50- 90 in the suggested new panel b. Otherwise, the reader is left to assume they are correctly identifying the features described in the text on their own. >> **Yes, thanks, adapted as suggested**

In lines 50-90, the "Geology of Urvara basin," several physiographic descriptions are provided about Urvara without always explaining the significance of these features. I suggest that for every physiographic description, the authors add a few words to tell the reader why this is pertinent to the high-impact results (finding of putative salts and organics). For example, there is a lot of text dedicated to presenting the smooth material in this section—I believe this is relevant because it implies that some resurfacing took place ~100 Myr after the crater's formation, and the area is relatively young. This detail should not be relegated to the Methods. Any initial discussion of the geology should explicitly frame the main results. >> **You are fully right. We hope that the restructured text and the additional information associated with the physiographic descriptions now enhance the understandability of the text.**

In lines 96-110, there is a brief description of the bright material on Ceres. Line 101 states: "These areas show mostly higher reflectances than the smooth material, often in the order of or exceeding the lower limit of BM on Ceres (0.035 at 0.55 μm)¹⁸"... Please add one sentence to explain the context of other BM on Ceres, where it has been discovered, and what it might be. For example, "BM discovered elsewhere on Ceres, is often associated with impact craters, but some instances exist where it is associated with mons or volcanic dome-like features¹⁸"). Good context is provided at the end of the section (line 121) for the very bright material. If the other reflectance units described in this section have analogs elsewhere on Ceres, please provide their context. >> **The text about the BM has been modified as you suggested. Context has been added for those color units, which are important for our study conclusions.**

In lines 194-201 contain a lot of detail that is not decipherable without referencing the Methods/Supplement. I suggest shortening this section and moving most of the text to the Methods, then distilling the essential results relevant to the paper's main findings. >> **Ok, thanks. Some details have been moved to the Methods section (robustness of model ages).**

Aside from the comments on the presentation described above, the work itself is convincing. Several previous color and spectral studies using the Dawn payload have been published and have demonstrated the validity of data returned by those instruments. The statistical analysis on crater ages appears thorough. I suggest where the authors should add additional details in my minor comments below.

Minor Comments

Line 99: Here, the term/unit Bright Material (BM) is mentioned for the first time. Please move up the quantitative definition of BM (currently on line 102) to the sentence where the unit is introduced. >> **Changed as suggested.**

Line 195: Above, I suggest much of this material be moved into the Methods. In the Methods, please explicitly define the criteria used for identifying pits versus craters. On line 986 (Supplement), I see the distinction that pits are "lacking raised rims", but any other criteria used during crater identification/counting should be explicitly stated in Methods. >> **Several lines have now been moved to the Methods section.**

Line 299: Exposures of (concentrated) BM showing higher reflectivity [insert reflectance threshold here] on floor and ejecta are rare. >> **This sentence isn't used anymore.**

Line 312: "The detection of organic material on a relatively young floor scarp let us conclude that organic material could be more widespread on Ceres than thought and in this case would be of endogenic origin and not due to exogenic infall."

I am not confident organic detection is not potentially exogenic in origin, given many recent findings of exogenous material on other asteroids (e.g. Tatsumi et al., 2021, DellaGiustina et al., 2021). Please add another sentence justifying this statement.

>> **The common understanding in the Dawn science team is today in favor of an endogenic origin of the organic material at Ernutet. In order to follow your comment, we expanded the text.**

Line 430: Please use the author's last name (Reddy) for the citation >> **Changed.**

Line 586/Figure 3: Please state the pixel scale or ground sample distance in the caption for the mosaic shown. >> **Pixel scale information added.**

Line 605/Figure 4: States "Error bars, $\pm 1.5\%$ (typical uncertainty)." Please provide the relevant calibration paper/reference and specify a relative (e.g., filter-to-filter) or absolute uncertainty. >> **A condensed section about the error determination has been added to the Methods section.**

Line 693: "presence of volatiles in the subsurface. Barlow and co-authors" >> **Typo corrected.**

Line 964: "The grooves are certainly not secondary crater chains, as we do not observe raised rims or ejecta deposits" >> **Typo corrected.**

Line 967: Supplementary Information Table 2: Crater Plots and Areas of Measurement. Many of the legends in these plots, and some of the plots themselves, are not legible. Please correct this in the next revision >> ***Figures are now enlarged.***

Reviewer #2 (Remarks to the Author):

General comments:

Although interesting, I find the paper not so easy to read because of the multiple back and forths between main text and the methods sections / extended data figures.

Important information on data analysis is given in the supplementary material, but shouldn't it be in the methods sections?

On the contrary, in the methods there are sections that are not « methodology » but give more description (central ridge cliffs, dark bluish material, nature of central ridge, rising brines): I do not understand the hierarchy between the different sections and kind of text/data/method/figures you present at the several places.

For example, in the section « crater based model ages » you cite ROIs number that are only defined lately in the extended data figures (#8 of 10): it is not easy to follow the main text while looking for important informations farther in the paper. Or, all the sites describes in the « color and spectral features section » has their spectra showed in another extended figure, but isn't it the main purpose of the paper?

>> You are completely right. The manuscript was originally written for and adapted to the style of Nature Astronomy and thus had to comply to a specific (lower) word limit for the main text. However, the updated version (updates in bold font in the manuscript) is now restructured to fit Nature Communications. We would like to thank you for your comments, suggestions, and the time you spent on the manuscript.

There is also some confusion between what is new content in this study and what has been published previously. Especially, line 58-59 you write « Descriptions of the geology and mineralogy of Urvara using lower resolution images can also be found elsewhere ». Thus, is the figure 1 as important to be put in the main text? The same for figure 3 (based on HAMO data), that is in a way presented in Crown et al. 2018 (your ref 12)?

>> The focus of the present manuscript is on the description and analysis of specific floor features of Urvara, which could be associated with deep-seated brines. Our analysis is based on high-resolution imagery (about 5 – 10 times higher than in the past), whose analysis hasn't been published so far. In order to provide the overall context, some previously published information is given too. We think this is quite usual. Figure 1 is now modified. Panel a shows the context (global surface in colors) of features discussed in the manuscript. Panel b, the clear filter mosaic of Urvara, is labelled with the features discussed within the text. We think that Figure 1 is mandatory to understand the context of the features we discuss and analyze.

You are right, a figure similar to Figure 3 has been published by Crown et al. 2018. The new figure (now Figure 4) displays a newly calibrated and processed false-colour mosaic of Urvara, with a considerably improved removal of residual photometric influence by topography. Those images used for Figure 4 exhibit a reduced uncertainty in reflectance compared to previously published mosaics and the photometric correction is enhanced by using the newer LAMO shape model. Thus, color differences are easier to identify than before. We think that such a color map is essential for the present manuscript, since it provides an overview about the color units we discuss.

Is the whole description in the section « Geology of Urvara basin » mandatory as it was described previously by other authors?

>> Yes, we think it is, since we show new features and discuss new aspects of floor features, which haven't and could not have been discussed before because of the non-available high-resolution imagery. At some passages the text has been shortened in consideration of your point with the issue of repetitions.

Also, for the crater-based model ages, some equivalent work has been done by Crown et al.: how do your study and results compare to this publication (your ref 12)?

>> The present study is based on the same method for deriving absolute model ages from crater size-frequency distributions like the study by Crown et al. (2018), which is a geologic mapping study for Urvara and Yalode, including respective ejecta blankets. Since our study is focused on Urvara only, it is more detailed in resolving variations in crater frequencies on the floor area of Urvara. At the end of chapter 6, Crown et al. (2018) qualitatively describe a similar observation which we point out in our manuscript, i.e. a smooth material region with lower crater frequency, implying a younger crater retention age on the floor of Urvara. In absolute model ages, there is also good agreement, because Crown et al. find ~190 Ma average age of the Urvara floor, while our measurements resolve the floor in roughly similar sized areas one older with ~260 Ma (crater formation) and one younger ~160 Ma (resurfacing age). In order to do so, we counted individual units of the floor material and found that it consists of older and younger areas, the older belong to the crater formation age, while the younger are likely associated to late floor activities (e.g. pitted terrain formation). Thus, the major finding, the youth of the smooth material, is in good agreement between both studies. In absolute model ages there is a difference, as our results consider much higher resolution images (up to 3 m/pixel instead of ~35 m/pixel), resulting in a better coverage of cratering effects, affecting only small craters and areas, respectively. Different measurement area mapping, crater identification abilities and even experience in fitting the resulting (sometimes noisy) crater distribution with a production function can easily produce certain differences in absolute model ages.

Another concern is that your paper title refers to mineralogical hints on Urvara, precisely brines and organics, but there is not that much about spectroscopy in here. The description of features is not sufficient to convince the reader: you use a reflectance level to conclude on the presence of salts, and similar colors to conclude on organics.

>> You are right, the focus is on the results obtained from the more recent high-resolution FC data and less on the VIR data. We used the high-resolution FC data to trigger new science. It is well known that the limited wavelength range of the FC does not allow unique mineralogic classifications. However, the organic-rich material at Ernutet and the salt deposits on Ceres exhibit typical spectral shapes in FC colors and thus can be used for a classification of these both lithologies. This is, for example, demonstrated in Thangjam et al. 2018 or when comparing identifications by Stein et al. 2019 with Palomba et al 2017. In order to verify the robustness of our FC based identification of organics, we used spectral data from VIR to check whether the characteristic spectral feature can be identified at this specific site. The presence of the spectral feature is confirmed in Figure 5 (now Fig. 8). Thus, we conclude on the presence of organics at this site in Urvara. Unfortunately, higher-resolution VIR data of this sites does not exist. Thus, we need to get along with the information we have.

Most of text and figures is related to geologic context that was already previously discussed.

>> Here we disagree. In the present manuscript several features of Urvara are discussed for the first time: e.g., CREs and central pit. Other features are described in more detail due to the higher resolution imagery available since the end of 2018 and the newly calibrated color mosaics in 2021. The updated and re-structured text highlights this.

Specific comments:

- line59: could you remind the reader what was the global mineralogy described by previous study for Urvara?

>> Because of the length of the manuscript we prefer to use here a reference and did not repeat published results on mineralogy.

- I recommend to not use parentheses for words that are incorporated in a sentence. E.g. l.105 « However, the (younger) smooth material » or l. 106 « The (dark) bluish craters ». It is not clear whether these words are hypothesis or important information. There are other occurrences throughout the text, please adopt another convention.

>> Those parentheses have been removed from the text.

- line215: you use the word « ammoniated » in the discussion/conclusion as a major constituent to explain geologic features, but this is not used before in the text and is included in fig.5 (NH₄) without explanation. Please clarify your spectroscopic detections (link with salts?).

>> We modified the respective figure, its caption, and the main text to improve the understandability of the spectroscopic detections. NH₄⁺ is found on most places at Ceres but less relevant to explain the discussed new findings.

- Fig2: I am not sure I understand what the « cross-axis » is here.

>> We mean the lateral axis, word changed.

- Fig3: I do not see the difference between the feature related to organics and the one to carbonates in the 3.4 microns region? Are the arrows misleading? Are they highlighting different components (you use « organics », « organic component »)?

>> The Urvara reddish BM, located on a young floor scarp, exhibits a pronounced 3.4 my m feature compared to sites outside the scarp (e.g., young smooth material). This pronounced 3.4 my m feature in combination with a weaker or absent carbonate feature at 3.9 my m is suggesting the presence of organics.

Furthermore, what are the spectral signatures around 3. 1 and 3.2 on the bottom spectrum (Cereal Tholus)?

>> According to De Sanctis et al. (2020) “the minima at 2.76 μm have been previously ascribed to phyllosilicates ... ; the minima at 2.88, 3.2 and 3.28 μm were observed in the previous data but not assigned to specific species”

- Fig5: Give in the caption the site number (e.g. red sloped terrain is #2)

>> **Site number now given in caption,**

Also, you mention « the BM site in fig.7 », but this should be « extended data figure 7 ».

>> **Corrected while restructured.**

There are no available VIR spectra for the other BM sites you present in ext. data fig. 6 to enrich the discussion?

>> **Many of the BM sites identified in Urvara, by using the FC data, are too small to be identified in VIR data. We think that a comparison between the Ernutet organic-rich material and the material on the floor scarp of Urvara is sufficient to show their resemblance. The Cerealia Tholus spectrum has been added to this figure to demonstrate its difference.**

- line636: should « orbit » be plural?

>> **Yes, thanks, corrected.**

- line643: should « These » be singular?

>> **Yes, thanks, corrected.**

- line 648: same for « reflectances »?

>> **Yes, thanks, corrected.**

- line667: I would define also here in the Methods section what the CREs acronym is (if the reader goes to this section while reading the main text in which the acronym is defined later).

>> **Due to the restructured text this comment isn't valid anymore.**

- line739-740: I do not understand the statement « ... to the Moon and Occator crater in Urvara the age differences occur within the floor of Urvara... ». Is there a coma missing after « crater »?

>> **Yes, thanks, the comma was missing.**

- ext. data fig2: What about including a ROI on this figure instead of the arrow? The arrow is hard to see and the ROI might define better the edge of the following figure 3.

>> **Yes, thanks, we added a polygon to highlight the pitted terrain. The projection of the figure has been changed to be complied with other figures of the manuscript.**

- ext. data fig3: Could you give the same longitude than on the previous figure2 (use E instead of W)? It could be easier for the reader to localize the region.

>> **Changed as suggested.**

- ext. data fig4: the x-axis is a bit weird, is there a reason that you give the numbers 99, 149, 198, and 248 instead of 100, 150, ...?

>> You are right, changed as suggested.

- ext. data fig5: Is it possible to put a scale on both views? And to separate them (a vertical white line or something) to see at first sight these are two distinct images (and not one view with some « blank » spaces on the FOV)?

>> Unfortunately scale bars are not possible due to the perspective view but approx. displayed sizes are given now in the caption. The sub-figures are now separated by a white line.

- ext. data fig6: I find these two plots hard to read (especially panel A because of all the plotted data). Could you add a number/letter/sign in front of each line (around 0.4 microns) to guide the eye?

>> Yes, thanks, numbers/information have been added.

- ext. data fig7: specify in the caption this is site #2 to remind the reader with previous cited sites.

>> Yes, thanks, information added to the caption.

- ext. data fig9: there is no explanation in the caption, is it intended? What about put this figure at the same scale than the previous one fig8 ? Or put the numbers from fig8 ROIs to correlate both figures?

Just a question (might be out of scope, my curiosity), but what could be the age of all the uncolored areas?

>> The scope of this figure is to illustrate the three different age ranges we found. Thus, we waived designations which are already given in (new) Fig. 10. Projections and grids are now harmonized. The caption is now a bit more detailed.

- ext. data fig10: It is hard to see the different regions you cite. Could you add some lines/arrow/signs to point towards what you describe in the caption?

>> Yes, thanks, we added this information to the figure.

REVIEWER COMMENTS

Reviewer #1 (Remarks to the Author):

Review #2 of "The Urvara basin on Ceres – brine residues and organics" by Nathues et al.

First, I thank the authors for considering and implementing my suggestions in their revised manuscript. I believe the manuscript, especially the figures, is much improved. The manuscript now stands on its own and is not bound to the methods for understandability. This effort to reorganize the manuscript is much appreciated.

I note that the newly implemented text from the authors did contain typos and odd phrasing of sentences or run-on sentences. I have made suggestions throughout the manuscript to fix these issues. Please refer to the marked-up manuscript I have submitted with this review. Line numbers referred to here are from that document. I have included several corrections and suggested text revisions for the authors to consider in that document. Still, the authors may benefit from a copy-editing service, as parts of the paper were challenging to parse. As a reviewer, my suggested edits are mainly intended to clarify scientific results.

I recognize the author's effort to make the manuscript more accessible to the broader community of scientists who read Nature Communications. Some areas of inaccessibility persisted in the revised draft, which I have tried to address directly in the marked-up version of the manuscript. For example, the authors refer to 'HAMO resolution' or 'LAMO resolution' throughout the text, but HAMO and LAMO mean little to those outside of the Dawn team; it is more cogent just to state the pixel scale of data (I have made this change throughout). I also made several small suggestions to help provide context when information is presented in the text for the first time, which may be unknown to a planetary scientist unfamiliar with Ceres.

I also found that the newly implemented text about color image uncertainties is insufficient (Lines 514-525). I suggest the authors expand on this to include more detailed information essential for anyone attempting to reproduce their findings. A brief description of the radiometric uncertainty limitations of the camera is merited. Please see my comments in the marked-up manuscript.

Finally, with the last major revision and reorganization, new issues stand out. I highlight them not to discourage the authors but because the more coherent revised manuscript makes them easier to recognize. Specifically, the discovery of salt and organics is the high-impact result that makes this paper deserving of publication in a Nature family journal—not the geology of the Urvara basin (as highlighted in the title). The paper should lead with this high-impact discovery and not the detailed description of Urvara, which is only relevant because it provides the geologic context for discovering brines and organics on Ceres. As such, I suggest the following structure of the paper:

1. Brief of Urvara to place it within the context on Ceres
2. Discussion of color/spectral measurements that indicate brines and organics (e.g., Distinctive color and spectral features section)
3. Detailed description of Urvara to place this discovery in geologic context and develop the hypothesis for the origin of the organic materials
4. Discussion

I believe this reorganization will significantly improve the manuscript's readability and place it in line with the structure of Nature-family papers (opposed to the more a journal like Icarus, where results are typically built up to later in the manuscript). The required elements are present in the paper, and this revision is only a reordering of those elements. I suggest that it be published after this reorganization, the copy edits I have indicated in the marked-up version of the manuscript, and more details on image radiometric uncertainties.

Reviewer #2 (Remarks to the Author):

The authors answered my concerns and questions.

The paper is now easier to read after being restructured.

New results have been emphasized/better explained (especially the spectral data).

I have tiny comments (below) and two final suggestions.

- Should the text in lines 204-215 come with a figure showing the results of your study on all ceraan craters?

I am not sure I follow the reasoning here: you find BM on craters larger than 25km (so a size criterion) but then you state that "the occurrence... seems to be rather independent from crater size". Could you please clarify?

- I am wondering if you should place the result section "crater-based model ages" before the "Distinctive colour and spectral features" section. In the "spectral" one you compare the younger and older SM" (l.185) but the reader does not know yet how this is determined...?

Once you describe the "geology" and the "age" of the terrains, then it's easier to deal with both to comment colours/spectra in their context.

Really just a suggestion...

Very minor comments:

- l.45: is there a typo on "thorough" that should be "through"?

- l.697: Vishnu Reddy first name is indicated in citation 57 instead of last name.

Response to reviewer #1

Review #2 of “The Urvara basin on Ceres – brine residues and organics” by Nathues et al.

First, I thank the authors for considering and implementing my suggestions in their revised manuscript. I believe the manuscript, especially the figures, is much improved. The manuscript now stands on its own and is not bound to the methods for understandability. This effort to reorganize the manuscript is much appreciated.

I note that the newly implemented text from the authors did contain typos and odd phrasing of sentences or run-on sentences. I have made suggestions throughout the manuscript to fix these issues. Please refer to the marked-up manuscript I have submitted with this review. Line numbers referred to here are from that document. I have included several corrections and suggested text revisions for the authors to consider in that document. Still, the authors may benefit from a copy-editing service, as parts of the paper were challenging to parse. As a reviewer, my suggested edits are mainly intended to clarify scientific results.

>> Thanks a lot for the detailed corrections and improvements you did on the text! The text is now copy edited as well, as you suggested. Changes are in bold letters.

I recognize the author's effort to make the manuscript more accessible to the broader community of scientists who read Nature Communications. Some areas of inaccessibility persisted in the revised draft, which I have tried to address directly in the marked-up version of the manuscript. For example, the authors refer to 'HAMO resolution' or 'LAMO resolution' throughout the text, but HAMO and LAMO mean little to those outside of the Dawn team; it is more cogent just to state the pixel scale of data (I have made this change throughout). I also made several small suggestions to help provide context when information is presented in the text for the first time, which may be unknown to a planetary scientist unfamiliar with Ceres.

>> Almost all of your suggestions are now implemented:

R11: Deleted as suggested.

R12: Changed as suggested.

R13: Added.

R14: Information added.

R15: Added as suggested.

R16: Changed as suggested.

R16: Changed a suggested.

R17: Rephrased as suggested.

R18: Changed a suggested.

R19: Added as suggested.

R110: Sentence removed since the information is not of high relevance here.

R111: Information added as suggested.

R112: Corrected.

R113: Changed as suggested.

R114: Text modified.

R115: Changed as suggested.

R116: Changed as suggested.

R117: Yes, the site was imaged more than once. A cosmic would have looked different.

R118/119: Text modified.

R120/121: The requested information has been added; the former section was moved to the “data processing” section and shortened.

I also found that the newly implemented text about color image uncertainties is insufficient (Lines 514-525). I suggest the authors expand on this to include more detailed information essential for anyone attempting to reproduce their findings. A brief description of the radiometric uncertainty limitations of the camera is merited. Please see my comments in the marked-up manuscript.

>> The calibration of the camera was a continuous process during which we improved its accuracy continuously. We added two references to the text, which describe the calibration. Further details can be found in the PDS and in project reports. The uncertainties were also continuously enhanced. Our approach is described in the further added references. We deleted the more general text of the former section “uncertainties of color spectra” and moved the new information to “data processing”. We believe that a more detailed description of the instrument would be too much for this (geologically) motivated paper.

Finally, with the last major revision and reorganization, new issues stand out. I highlight them not to discourage the authors but because the more coherent revised manuscript makes them easier to recognize. Specifically, the discovery of salt and organics is the high-impact result that makes this paper deserving of publication in a Nature family journal—not the geology of the Urvara basin (as highlighted in the title).

The paper should lead with this high-impact discovery and not the detailed description of Urvara, which is only relevant because it provides the geologic context for discovering brines and organics on Ceres. As such, I suggest the following structure of the paper:

1. Brief of Urvara to place it within the context on Ceres
2. Discussion of color/spectral measurements that indicate brines and organics (e.g., Distinctive color and spectral features section)
3. Detailed description of Urvara to place this discovery in geologic context and develop the hypothesis

for the origin of the organic materials

4. Discussion

>> We have had an extensive discussion among the authors and decided to adhere to the current structure. The primary reason for this is that the geologic description also contains new results based on the more recent and highest resolution XM2 data. Equally important is that the geologic context is essential to understand the main findings that follow. Restructuring the manuscript as recommended would disrupt the sequence of things and leave the reader initially without the geologic context and finally would increase the number of words required. We went through a number of published papers in Nature Communications and found that the present structure is common and in accordance with the style guidelines. Hence, we believe that the current structure is justified, and that the used “standard structure” is proper for this manuscript.

I believe this reorganization will significantly improve the manuscript's readability and place it in line with the structure of Nature-family papers (opposed to the more a journal like Icarus, where results are typically built up to later in the manuscript). The required elements are present in the paper, and this revision is only a reordering of those elements. I suggest that it be published after this reorganization, the copy edits I have indicated in the marked-up version of the manuscript, and more details on image radiometric uncertainties.

Response to reviewer #2

Reviewer #2 (Remarks to the Author):

The authors answered my concerns and questions.

The paper is now easier to read after being restructured.

New results have been emphasized/better explained (especially the spectral data).

>> Once again thanks for your time. We have modified the text (changes are in bold) during the actual revision to improve its readability.

I have tiny comments (below) and two final suggestions.

- Should the text in lines 204-215 come with a figure showing the results of your study on all cerean craters?

>> These more detailed results are planned for a later publication.

I am not sure I follow the reasoning here: you find BM on craters larger than 25km (so a size criterion) but then you state that "the occurrence... seems to be rather independent from crater size". Could you please clarify?

>> You are right, the sentence is now modified.

- I am wondering if you should place the result section "crater-based model ages" before the "Distinctive colour and spectral features" section. In the "spectral" one you compare the younger and older SM" (l.185) but the reader does not know yet how this is determined...?

Once you describe the "geology" and the "age" of the terrains, then it's easier to deal with both to comment colours/spectra in their context.

Really just a suggestion...

>> This is a valid option. However, we prefer to add a cross-reference to overcome the missing information in the color section.

Very minor comments:

- l.45: is there a typo on "thorough" that should be "through"?

- l.697: Vishnu Reddy first name is indicated in citation 57 instead of last name.

>> Thanks, corrected.